# Differences between Tuberous Sclerosis Complex Patients with and without Epilepsy: The Results of a Quantitative Diffusion Tensor Imaging Study

**DOI:** 10.3390/biomedicines12092061

**Published:** 2024-09-10

**Authors:** Anna B. Marcinkowska, Sergiusz Jóźwiak, Agnieszka Sabisz, Agnieszka Tarasewicz, Beata Rutkowska, Alicja Dębska-Ślizień, Edyta Szurowska

**Affiliations:** 1Applied Cognitive Neuroscience Lab, Department of Neurophysiology, Neuropsychology and Neuroinformatics, Medical University of Gdańsk, Tuwima Str. 15, 80-210 Gdańsk, Poland; 22nd Department of Radiology, Medical University of Gdańsk, 80-210 Gdańsk, Poland; asabisz@gumed.edu.pl (A.S.); bea_rut@wp.pl (B.R.); eszurowska@gumed.edu.pl (E.S.); 3Research Department, The Children’s Memorial Health Institute, 04-730 Warsaw, Poland; sergiusz.jozwiak@gmail.com; 4Department of Nephrology Transplantology and Internal Diseases, Medical University of Gdańsk, 80-210 Gdańsk, Poland; ataras@gumed.edu.pl (A.T.); adeb@gumed.edu.pl (A.D.-Ś.)

**Keywords:** tuberous sclerosis complex, diffusion tensor imaging

## Abstract

Introduction: Tuberous sclerosis complex (TSC) is a neurocutaneous disease with a high incidence of epilepsy and damaging effects on cognitive development. To understand the mechanisms leading to abnormal cognitive development, diffusion tensor imaging (DTI) techniques have begun to be used in recent years. The present study is the first to investigate differences in the microstructure and integrity of white matter tracts in adult patients with TSC and with and without epilepsy. Method: A total of 37 patients with TSC (18 with epilepsy, median age 36 years; 19 without epilepsy, median age 35 years) without intellectual disability and autism spectrum disorder were included in the study. The control group (median age 34 years) comprised 37 individuals without psychiatric or neurodevelopmental disorders and neurological and cardiovascular diseases, diabetes, or addictions. A magnetic resonance imaging (MRI) DTI sequence was applied. Results: There were differences in the average values of DTI parameters between patients with TSC and epilepsy and patients with TSC but without epilepsy in five white matter bands. When comparing the average values of DTI parameters between patients with TSC and epilepsy and healthy controls, we found differences in 15 of 20 analysed white matter fibres. White matter tracts in patients with TSC and epilepsy had more abnormalities than in patients with TSC but without epilepsy. The former group presented abnormalities in longer white matter fibres, especially in the left hemisphere. However, the latter group presented abnormalities in more medial and shorter white matter fibres. Conclusion: This DTI study documents the changes in the brain white matter of patients with TSC associated with the presence of epilepsy.

## 1. Introduction

Tuberous sclerosis complex (TSC) is a rare multisystem neurocutaneous disorder that is associated with the formation of hamartomatous lesions in various organs, including the brain [1]. Conventional magnetic resonance imaging (MRI) in TSC has revealed three types of brain lesions: cortical tubers (present in 90% of individuals with TSC), subependymal nodules (SENs; present in 80% of individuals with TSC), and subependymal giant cell astrocytomas (SEGAs; present in up to 15% of individuals with TSC) [2,3]. Further histological examinations of these brain hamartomas have revealed characteristics of aberrant neuronal and astrocyte expansion and cell enlargement such as microdysgenesis and abnormal cortical layering [4]. Four unique patterns can be found in MRI of white matter (WM) lesions: (1) wedge-shaped lesions, (2) non-specific conglomerate foci, (3) straight or curved radial bands extending from the ventricle through the cerebral WM toward the cortex, and (4) cerebellar radial bands [5,6,7]. One of the limitations of conventional MRI is the inability to obtain the microstructural characterisation of tissues, including abnormal differentiation, migration, organisation, myelination, and structural connectivity.

Changes in the structure of the brain, in turn, give rise to the clinical manifestations of epilepsy, which is the most prevalent neurological symptom in patients with TSC and affects 70–90% of patients [8,9]. There are different types of convulsions, such as focal, multifocal, infantile, or a combination of these and others. Infantile spasms can occur in up to 30% of people affected by TSC. People with TSC frequently have multiple seizures that do not improve with medication [10]. Epileptogenic foci can occur in different parts of the brain at different times. In longitudinal studies, changes in the morphology of the seizures have been observed in more than half of the patients. The deficiency of GABAergic interneurons explains the early onset and severe course of TSC-related seizures; thus, the GABA transaminase inhibitor vigabatrin is proven to be effective in approximately 95% of patients [11]. Other anti-epileptic drugs are less effective in TSC. Some researchers suggest that white matter disruption is related with epilepsy in TSC [12].

Diffusion tensor imaging (DTI) is a method that uses MRI’s capacity to assess the direction and amount of water diffusion in tissues while they are still in motion [13]. This provides a view of microscopic tissue architecture and is used to obtain information about the value and direction of the diffusion of hydrogen ions of water molecules, which allows for a broader diagnosis of changes in the microstructure of the brain in vivo. Several DTI parameters reflect the integrity of the WM, including the fractional anisotropy coefficient (FA), a measure of diffusion directionality; radial diffusivity (RD) and axial diffusivity (AD), which are indicators of myelin degeneration and of axonal damage [14], respectively; and mean diffusivity (MD), which determines the general properties of the tissue [15]. More precisely, FA is interpreted as a quantitative biomarker of WM ‘integrity’. A confirmation for this interpretation is the fact that pathological studies tend to show a reduction in FA associated with neurogenerative processes [16,17]. RD is often more sensitive to myelin changes; however, an increase in RD can indicate myelin loss, axonal loss, or decreased axonal packing density [18]. AD is a measure of the extent of diffusion along the direction of fibre tracts. A decrease in AD can indicate axonal damage, reduced axonal thickness, or less organised axonal orientation. Research suggests that AD is unaffected by myelin [19]. Finally, an overall increase in MD generally indicates an increase in water content (such as from oedema and inflammation), leading to reduced resistance and, consequently, higher diffusion rates [20].

The initial studies showing impaired diffusion in individuals with TSC focused on subcortical tubers. They revealed reduced FA and increased MD, as confirmed by histopathological findings of tuber tissue (dysmorphic, dysplastic, disorganised, and enlarged cells) [21]. However, abnormalities visualised by DTI are not limited to the WM surrounding tubers [22,23], as abnormalities in myelination, migration, and neuronal differentiation go beyond discrete borders of tubers visible on conventional MRI. Normal-appearing white matter (NAWM) refers to areas around WM lesions (e.g., cortico–subcortical tubers) that appear normal with conventional MRI but may show low perfusion or microstructural changes [24,25,26]. This has important implications because in addition to multifocal neuropathology in the form of tubers, these abnormalities reflect a global or diffuse microstructural pathology of WM, also confirmed in neuropathological studies of TSC [23]. An increasing number of researchers are using DTI to describe WM abnormalities that appear unaltered on conventional imaging [27,28,29,30,31,32].

Typically, DTI studies have included small or mixed groups (children and adults, with and without epilepsy), which is not sufficient to obtain detailed data on microstructural changes specific to clinically different patients [30,32,33,34,35]. In the present study, we evaluated WM microstructure and structural connections of the brain in high-functioning (without autism spectrum disorder and intellectual disability) adults with TSC, depending on the presence or absence of epilepsy. To date, no conventional imaging biomarker reliably relates to this neurophenotype. Our study is the first to use magnetic resonance tractography to determine the characteristics of the diffusion coefficients of NAWM fibres in the brain of patients with TSC, comparing individuals with and without epilepsy.

## 2. Materials and Methods

This study protocol was approved by the Ethics Committee of the Medical University of Gdańsk (NKEBN/682/2018-2019). All participants and/or caregivers were informed about the study objectives and procedure and signed a written consent form.

### 2.1. Participants

All individuals referred to the national TSC reference clinical centre at the Department of Nephrology, Transplantology, and Internal Medicine underwent a detailed analysis of medical records, a physical examination performed by a nephrologist and/or an internal medicine specialist, an assessment of skin lesions and oral manifestations, kidney and brain MRI, and high-resolution lung computed tomography. One-hundred subjects who met the criteria for a diagnosis of TSC based on the international diagnostic guidelines [36] were examined by a clinical psychologist. First qualification for the project was carried out using the TAND Checklist [37]. Further detailed psychological assessment involved mood and cognitive testing as well as the assessment of neurodevelopmental disorders [38] for assessing individuals without ASD and intellectual disability. A schematic illustration of the patient enrolment process is presented in Figure 1.

Thirty-seven patients with TSC who met the following criteria were included in the study: (1) no intellectual disability, (2) no ASD, and (3) no contraindications for MRI examination. They were divided into two groups: one group based on the presence or previous medical history of epileptic seizures (EpiTSC, n = 18; median age 36 years) and another group without epilepsy (NEpiTSC, n = 19; median age 35 years). The history of epilepsy was established based on a review of clinical records. The control group of healthy volunteers was to match the age and education level of the TSC group. It comprised 37 individuals (median age 34 years) fulfilling the following criteria: (1) no psychiatric or neurodevelopmental disorders; (2) no neurological or cardiovascular diseases, diabetes, or addictions; and (3) no contraindications for MRI. Demographic variables did not differ significantly between the groups.

### 2.2. MRI Acquisition

Neuroimaging was performed with a Philips Achieva 3.0T TX MRI scanner (Philips Healthcare, Best, The Netherlands) using a 32-channel head coil. The study protocol consisted of a routine brain scan involving T2-weighted turbo spin echo imaging, a T1-weighted turbo field echo, and fluid attenuated inversion recovery (FLAIR) in three acquisition planes. T1-weighted post-contrast imaging was performed in the sagittal plane with a 1 × 1 × 1 mm voxel size (T1-TFE: repetition time [TR] = 7.44 ms, echo time [TE] = 3.6 ms, slice thickness = 1 mm, matrix = 260 × 240, and field of view [FOV] = 260 × 240 mm). The FLAIR sequence was applied in the transverse plane positioned parallel to the genu and splenium of the corpus callosum (TR = 9000–11,000 ms, TE = 125 ms, slice thickness = 4 mm, gap = 1 mm, and the matrix and FOV were adapted to the patient’s head). In addition to the structural examination, DTI was performed. The DTI sequence parameters were as follows: b-factor = 0 s/mm^2^ and 800 s/mm^2^ with 32 gradient directions, TR = 6900 ms, TE = 65 ms, voxel size = 2 × 2 × 2 mm, FOV = 230 × 230 mm, 70 slices, and the number of signals averaged = 1.

#### Image Analysis

The DTI sequence was acquired axially without angulation. T1-weighted images were converted to nii format using MRIConvert. The DTI images were estimated for the tensor model using the REKINDLE algorithm in the ExploreDTI software (Image Sciences Institute–University Medical Center Utrecht, The Netherlands, https://www.exploredti.com/generalinfo.htm, accessed on 6 May 2020); then, the images were corrected for motion and eddy current distortion, and the B matrixes were rotated. Further, non-brain tissues were removed from the images using the BET FSL tool (FSL 6.0, FMRIB, Oxford, UK). The processed data were used to calculate diffusion maps (FA, MD, RD, and AD) in the DTIFIT FSL software (FSL 6.0, FMRIB, Oxford, UK). The FA images of all patients were aligned to the standard FA image with a resolution of 1 × 1 × 1 mm (FMRIB58_FA). The probabilistic atlas of WM tractography of John Hopkins University was used to draw the neural pathways. The prepared tracts were plotted on the FA, MD, RD, and AD images of each patient and were used to calculate the mean value in each pathway. The obtained results were used to calculate the descriptive statistics and the intergroup analysis. Ten hemisphere-specific (three projection fibres and seven associative fibres) and two commissural pathways were identified, meaning that there were twenty paths in the final analyses.

### 2.3. Statistical Analysis

SPSS Statistics for Windows Version 27 (IBM, Corp., Armonk, NY, USA) was used for data analysis. The quantitative data are presented as the mean ± standard deviation (SD), and the qualitative data are presented as the absolute count and frequency. The normality of the distribution of each quantitative variable was assessed with the Shapiro–Wilk test. Student’s *t*-test was used to analyse normally distributed variables, and the Mann–Whitney U-test was used to analyse non-normally distributed variables. In addition, the four DTI parameters of the analysed NAWM fibres were compared between the three groups (EpiTSC, NEpiTSC, and control). For individual trials and tests, the normality of the distribution was checked; then, the data were compared using Student’s *t*-test (normally distributed data) or the Mann–Whitney U-test (non-normally distributed data). The Bonferroni correction was applied to the analysed data to account for multiple comparisons. Thus, *p* ≤ 0.0025 was considered to indicate a statistically significant difference.

## 3. Results

Two independent observers assessed the WM fibres based on MRI scans in the FLAIR sequence. Kendall’s coefficient of concordance confirmed agreement between the observers (W = 0.93). In each patient with TSC, each observer decided, independently, whether the 20 analysed WM fibres were NAWM or whether there were structural abnormalities. The number of fibres marked as NAWM in the groups is presented in Table 1. In the EpiTSC and NEpiTSC groups, the observers noted microstructural differences in NAWM that could be visualised by DTI. Moreover, there were significant differences between these groups. Myelin and axonal abnormalities were more pronounced in the EpiTSC group. In the EpiTSC and NEpiTSC groups, fibres linking distant lobes of the brain, especially frontal circuits, were disrupted.

When comparing the average values of the DTI parameters for the EpiTSC and NEpiTSC groups, there were significant differences in five NAWM bands. Specifically, there were MD differences in three WM pathways: the left inferior fronto-occipital fasciculus (IFOF) and the left and right inferior longitudinal fasciculus (ILF). Two NAWM pathways showed higher MD and lower FA in the EpiTSC group compared with the NEpiTSC group. There were significant differences in RD only in the left upper longitudinal bundle and for AD only in the left IFOF. Table 2 provides detailed data on significant differences in the diffusion parameters of NAWM for the EpiTSC and NEpiTSC groups.

When comparing the DTI results between the individuals with TSC and healthy subjects, we observed more microstructural changes within the NAWM in the EpiTSC group compared with the NEpiTSC group. In the EpiTSC group, the diffusion coefficients for the majority of the analysed NAWM fibres differed from the control group. In the EpiTSC group, the most significant differences were related to MD. When comparing the average values of the DTI parameters between the EpiTSC and control groups, there were significant differences for 15 of the 20 analysed WM fibres. MD was higher in 12 NAWM fibres in the EpiTSC group compared with the control group. There were similar differences in four and five WM tracts for AD and RD, respectively. FA significantly differentiated six NAWM pathways between the EpiTSC and control groups. The average FA for the EpiTSC group was lower than the NEpiTSC and control groups. All DTI parameters were significantly different in the left anterior thalamic radiation and the left superior longitudinal fasciculus. There were also significant differences in RD, FA, and MD in the left uncinate fasciculus. In the remaining NAWM pathways, one or two diffusion coefficients differed significantly between the EpiTSC and control groups. Table 3 provides detailed data on significant differences in the diffusion parameters of the NAWM fibre bands between the EpiTSC and control groups.

When comparing the NEpiTSC and control groups, we noted significant differences in the diffusion coefficients for six NAWM fibres. Two NAWM fibres—the right cingulum and forceps minor—showed higher MD in the NEpiTSC group. RD and AD were higher in four and three NAWM fibres, respectively, in the NEpiTSC group compared with the control group. However, FA was significantly lower in only one of the analysed NAWM fibres in the NEpiTSC group compared with the control group. None of the analysed fibres differed significantly in terms of all four DTI indices. Only the forceps minor (RD, MD, and AD) and right cingulum (cingulate gyrus; RD, FA, and MD) differed significantly in three analysed parameters. Similarly to the EpiTSC group, there were differences in RD, indicating myelin damage, in the forceps minor, the right anterior thalamic radiation, and the right cingulum. We also noted differences between the NEpiTSC and control groups in one left-hemisphere tract: the IFOF. There were differences in AD, indicative of axonal damage, in the forceps minor and the left cingulum, the corticospinal tract, and the superior longitudinal fasciculus (temporal part). FA was significantly lower in the right cingulum of the NEpiTSC group compared with the control group. None of the analysed fibres differed significantly between the groups in terms of all four DTI indices. The most abnormal fibres compared with the healthy group were the lesser forceps (higher RD, MD, and AD) and the right cingulum (higher RD and MD, and lower FA), each of which showed significant differences in three DTI parameters. The NEpiTSC group presented higher RD and AD in four WM fibres. Table 4 presents detailed data on significant differences in the diffusion parameters of the NAWM fibres between the NEpiTSC and control groups.

## 4. Discussion

We aimed to determine the characteristics of the DTI parameters of NAWM fibres in the brains of high-functioning individuals with TSC, comparing those with and without epilepsy. Our three main findings are as follows: (1) NAWM shows more abnormalities in individuals with TSC and epilepsy than in individuals with TSC but without epilepsy; (2) in individuals with TSC and epilepsy, NAWM abnormalities are present in longer WM fibres, especially in the left hemisphere; and (3) individuals with TSC but without epilepsy present NAWM abnormalities in more medial and shorter WM fibres compared with individuals with TSC and epilepsy. These findings indicate more severe abnormalities in the NAWM of individuals with TSC and epilepsy than in individuals with TSC but without epilepsy. These abnormalities are present especially in long WM fibres and refer to both axonal and myelin damage.

Despite the fact that the centrifugal pattern of WM diffusion abnormalities likely indicates astrogliosis and microstructural disruption associated with seizure activity near the focus [39], research has highlighted the epilepsy burden in the ipsilateral WM distant networks. Campos et al. [40] evaluated WM changes in focal epileptic lesions not related to TSC. They found that the variable distribution of WM damage in these patients indicates that the location of epileptic networks might influence the WM burden and that the underlying cause of epilepsy is likely an additional factor that contributes to this WM damage. Furthermore, Yasuda et al. [41] found differences in the FA of all 10 tracts in individuals with temporal lobe epilepsy versus controls. Thus, the existence of more severe WM abnormalities in patients with TSC and epilepsy relative to the cognitive profile of individuals with TSC but not epilepsy is not surprising considering the deteriorating effect of epileptic discharges on the structure of the brain [42]. These findings might explain the behavioural, cognitive, and affective problems in individuals with TSC, a conclusion also drawn by Sato et al. [33].

Researchers have claimed that the disruption of WM integrity can lead to reduced connectivity between cortical and subcortical brain regions. Our results are consistent with previous studies: we mostly found differences in directionality in individuals with TSC and epilepsy compared with healthy controls in long WM fibres of the left hemisphere. The observed changes in individuals with TSC and epilepsy in AD and RD may indicate axonal and myelin damage, respectively [19]. In the case of individuals with TSC and epilepsy, it is difficult to rule out WM damage caused by seizures.

The individuals with TSC and epilepsy showed differences in medial and short WM fibres compared with the healthy controls. These results indicate that epilepsy is not the only cause of WM disruption in TSC. In the individuals with TSC but without epilepsy, AD and RD, rather than FA, showed the most pronounced differences, indicating that both myelin and axonal damage are present in the NAWM fibres in those individuals. Surprisingly, the integrity of NAWM fibres is less affected than other microstructural aspects. Those results suggest an underlying biological process in TSC WM alternations. Consistently, Gruper et al. [43] examined patient-derived primary oligodendroglial cells and TSC2 knockout cells and concluded that oligodendroglial maturation and proper myelin sheath production are hindered due to disturbances in the mammalian target of the rapamycin (mTOR) pathway. Therefore, the oligodendroglial pathology may represent a direct consequence of an abnormal genetic program rather than merely being an inactive byproduct of chronic epilepsy [44].

Our findings indicate that in individuals with TSC, regardless of whether they also have epilepsy, there are microstructural differences in NAWM that can be visualised by DTI. The quantification of WM alterations by advanced diffusion MRI may be an additional biomarker for TSC and may be advantageous as a complementary MR protocol for the evaluation of that group of patients. Further research should focus on the analysis of DTI parameters in comparison to cognitive phenotypes and extend the diagnostics to include myelin networks. These studies could provide important information to understand the pathomechanism and to identify specific phenotypes of patients in whom effective therapies could be adapted to their structural and functional disorders.

## 5. Conclusions

This study represents the first attempt to use magnetic resonance tractography to characterise the diffusion coefficients of NAWM fibres in the brains of individuals with TSC and with or without epilepsy. A crucial aspect of this study was our comparison of the mean values of DTI parameters between patient groups in an attempt to elucidate the underlying causes of different clinical phenotypes in individuals with TSC. The analysis revealed more pronounced myelin and axonal abnormalities in individuals with TSC and epilepsy. Additionally, we observed most severe alternations in fibres connecting distant brain lobes in individuals with TSC and epilepsy. Furthermore, these individuals presented NAWM abnormalities in longer WM fibres, especially in the left hemisphere. These findings could potentially shed light on the behavioural, cognitive, and affective issues experienced by individuals with TSC. Future research should delve into comparing DTI parameters with cognitive phenotypes and expanding diagnostics to include myelin networks. Such investigations hold promise for providing valuable insights into the pathomechanisms underlying TSC and identifying specific patient phenotypes. Ultimately, this could lead to the development of tailored therapies that address both structural and functional abnormalities in affected individuals.

## Figures and Tables

**Figure 1 biomedicines-12-02061-f001:**
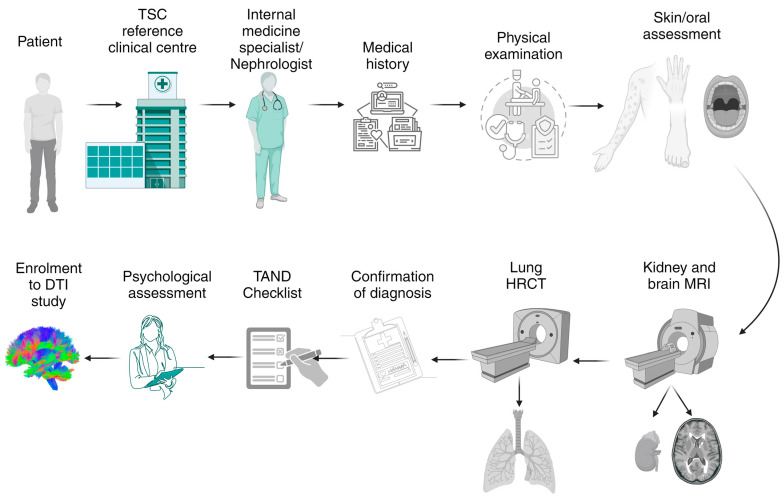
Schematic illustration of the detection methods used in studying TSC onset in this study. Before enrolment in the diffusion tensor imaging (DTI) study, all patients referred to the TSC reference clinical centre underwent a detailed assessment performed by a nephrologist and/or an internal medicine specialist. Medical records were analysed in case of epilepsy in the past and current onsets. Each patient underwent skin and oral assessment for lesions and clinical manifestations of TSC. All patients had kidney and brain magnetic resonance imaging (MRI) and high-resolution computed tomography (HRCT) of lungs. Patients who met the criteria of TSC were referred for a clinical psychologist examination. All patients were assessed with the TAND Checklist. A more detailed psychological examination included cognitive and mood testing as well as symptoms of neurodevelopmental disorders. (Created in BioRender.)

**Table 1 biomedicines-12-02061-t001:** NAWM fibres included in the EpiTSC (n = 17) and NEpiTSC (n = 19) groups.

White Matter Fiber	EpiTSC	NEpiTSC
Anterior Thalamic Radiation L	7	16
Anterior Thalamic Radiation R	10	18
Cingulum (Cingulate Gyrus) L	10	14
Cingulum (Cingulate Gyrus) R	9	18
Cingulum (Hippocampus) L	13	16
Cingulum (Hippocampus) R	14	16
Corticospinal Tract L	12	16
Corticospinal Tract R	12	17
Forceps Major	12	16
Forceps Minor	9	9
Inferior Fronto-Occipital Fasciculus L	4	17
Inferior Fronto-Occipital Fasciculus R	5	12
Inferior Longitudinal Fasciculus L	4	6
Inferior Longitudinal Fasciculus R	5	7
Superior Longitudinal Fasciculus L	4	13
Superior Longitudinal Fasciculus R	4	7
Superior Longitudinal Fasciculus (Temporal Part) L	3	9
Superior Longitudinal Fasciculus (Temporal Part) R	3	8
Uncinate Fasciculus L	7	13
Uncinate Fasciculus R	14	18

**Table 2 biomedicines-12-02061-t002:** Statistically significant differences in the DTI parameters of NAWM fibres between the EpiTSC and NEpiTSC groups.

WM Fiber	DTI	Mean EpiTSC	SD	Mean NEpiTSC	SD	*t*/U	*p*
Cingulum (Cingulate Gyrus) R	FA	0.42110	0.03376	0.41964	0.02777	21.00	0.002
IFOF L	MD	0.00084	0.00016	0.00083	0.00005	1.00	0.002
	AD	0.00134	0.00008	0.00117	0.00006	0.00	0.0001
ILF L	FA	0.32371	0.01998	0.36734	0.01316	0.00	0.002
	MD	0.00083	0.00016	0.00082	0.00005	02.00	0.0019
ILF R	MD	0.00080	0.00015	0.00079	0.00005	3.00	0.0021
SLF (Temporal Part) L	RD	0.00076	0.00007	0.00068	0.00004	0.000	0.002

L—left, R—right, WM—white matter, SD—standard deviation, ATR—anterior thalamic radiation, IFOF—inferior fronto-occipital fasciculus, ILF—inferior longitudinal fasciculus, SLF—superior longitudinal fasciculus, DTI—diffusion tensor imaging coefficient, MD—mean diffusivity, FA—fractional diffusivity, RD—radial diffusivity, AD—axial diffusivity.

**Table 3 biomedicines-12-02061-t003:** Statistically significant differences in the DTI parameters of NAWM fibres between the control group and the EpiTSC group.

WM Fiber	DTI	Mean Control	SD	Mean EpiTSC	SD	*t*/U	*p*
ART L	MD	0.00086	0.00009	0.00101	0.00026	127.5	0.0001
ART R	RD	0.00067	0.00009	0.00085	0.00033	76.500	0.002
	FA	0.37941	0.04454	0.34572	0.09183	79.00	0.002
	MD	0.00084	0.00009	0.00099	0.00029	131.00	0.0001
	AD	0.00118	0.00010	0.00129	0.00033	74.50	0.002
Cingulum (Cingulate Gyrus) L	RD	0.00057	0.00009	0.00064	0.00016	50.00	0.001
Cingulum (Cingulate Gyrus) R	FA	0.41279	0.05245	0.33500	0.03376	6.00	0.0001
	MD	0.00074	0.00007	0.00082	0.00016	136.50	0.0001
Cingulum (Hippocampus) R	FA	0.32706	0.04146	0.31413	0.07636	116.00	0.001
	AD	0.00114	0.00012	0.00113	0.00014	102.5	0.001
Corticospinal Tract L	MD	0.00076	0.00007	0.00079	0.00015	162.00	0.002
	AD	0.00119	0.00008	0.00107	0.00013	58.500	0.0001
Forceps Major	FA	0.44912	0.03987	0.39673	0.02931	12.00	0.002
Forceps Minor	MD	0.00066	0.00007	0.00073	0.00016	114.5	0.0001
	AD	0.00098	0.00007	0.00139	0.00015	57.50	0.002
IFOF L	MD	0.00079	0.00007	0.00084	0.00016	116.5	0.0001
ILF L	MD	0.00079	0.00008	0.00083	0.00016	142.00	0.001
SLF L	RD	0.00064	0.00007	0.00082	0.00028	1.000	0.001
	FA	0.32648	0.03857	0.27432	0.03893	14.00	0.002
	MD	0.00078	0.00007	0.00083	0.00015	100.50	0.0001
	AD	0.00105	0.00008	0.00119	0.00007	6.00	0.001
SLF R	MD	0.00078	0.00008	0.00082	0.00015	154.00	0.001
SLF (Temporal Part) L	MD	0.00078	0.00007	0.00083	0.00015	106.00	0.0001
SLF (Temporal Part) R	MD	0.00079	0.00008	0.00082	0.00015	156.00	0.001
Uncinate Fasciculus L	RD	0.00061	0.00008	0.00068	0.00017	62.00	0.002
	FA	0.37255	0.04281	0.34928	0.08677	53.00	0.002
	MD	0.00078	0.00008	0.00081	0.00016	157.50	0.002

L—left, R—right, WM—white matter, SD—standard deviation, ATR—anterior thalamic radiation, IFOF—inferior fronto-occipital fasciculus, ILF—inferior longitudinal fasciculus, SLF—superior longitudinal fasciculus, DTI—diffusion tensor imaging coefficient, MD—mean diffusivity, FA—fractional diffusivity, RD—radial diffusivity, AD—axial diffusivity.

**Table 4 biomedicines-12-02061-t004:** Statistically significant differences in the DTI parameters of NAWM fibres between the control group and the NEpiTSC group.

WM Fiber	DTI	Mean Control	SD	Mean NEpiTSC	SD	*t*/U	*p*
ATR R	RD	0.00067	0.00009	0.00079	0.0029	166.00	0.002
Cingulum (Cingulate Gyrus) L	AD	0.00114	0.00008	0.00115	0.00005	73.00	0.0001
Cingulum (Cingulate Gyrus) R	RD	0.00063	0.00008	0.00065	0.00011	136.00	0.0001
	FA	0.41279	0.05255	0.37680	0.02777	129.00	0.001
	MD	0.00074	0.00008	0.00080	0.00005	161.50	0.002
Corticospinal Tract L	AD	0.00119	0.00008	0.00124	0.00005	134.00	0.002
Forceps Minor	RD	0.00050	0.00007	0.00100	0.00207	163.00	0.002
	MD	0.00066	0.00007	0.00091	0.00004	145.00	0.002
	AD	0.00098	0.00007	0.00105	0.00005	114.00	0.0001
SLF L	AD	0.00111	0.00008	0.00117	0.00004	123.00	0.002

L—left, R—right, WM—white matter, SD—standard deviation, ATR—anterior thalamic radiation, SLF—superior longitudinal fasciculus, DTI—diffusion tensor imaging coefficient, MD—mean diffusivity, FA—fractional diffusivity, RD—radial diffusivity, AD—axial diffusivity.

## Data Availability

The raw data supporting the conclusions of this article will be made available by the authors on request.

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
