# Peer review of "Differences between Tuberous Sclerosis Complex Patients with and without Epilepsy: The Results of a Quantitative Diffusion Tensor Imaging Study"

_biomedicines, 2024, doi:10.3390/biomedicines12092061_

Round 1
Reviewer 1 Report
Comments and Suggestions for Authors
In this manuscript, “Differences Between Tuberous Sclerosis Complex Patients With and Without Epilepsy: The Results of a Quantitative Diffusion Tensor Imaging Study”, authors analysed the differences in Tuberous Sclerosis patients with and without Epilepsy. Though the study represent limited results, the use of MRT, the number of patients with different group and statistical analysis reflect the weightage of the study. The results also go well with the title and the conclusion. Moreover, the discussion is represented with sufficient references. So, I recommend this manuscript for publication in MDPI Biomedicines.
Author Response
Dear Reviewer,
In this manuscript, “Differences Between Tuberous Sclerosis Complex Patients With and Without Epilepsy: The Results of a Quantitative Diffusion Tensor Imaging Study”, authors analysed the differences in Tuberous Sclerosis patients with and without Epilepsy. Though the study represent limited results, the use of MRT, the number of patients with different group and statistical analysis reflect the weightage of the study. The results also go well with the title and the conclusion. Moreover, the discussion is represented with sufficient references. So, I recommend this manuscript for publication in MDPI Biomedicines.
Response: Thank you very much for taking the time to review this manuscript. Thank you for your kind words.
Kind regards,
Anna Marcinkowska
Reviewer 2 Report
Comments and Suggestions for Authors
This manuscript employs DTI techniques to characterize differences in brain microstructures among patients with and without epilepsy, as well as in healthy controls. Statistical analyses of DTI parameters across these three groups have highlighted distinctive patterns in regions with impaired white matter. The manuscript is well-written and presents its findings clearly. The manuscript appears to require only minor revisions. To enhance readability, I recommend providing a more detailed description of TSC with epilepsy in the introduction. Specifically, it would be helpful to introduce the symptoms, patient population and summarize current research on the mechanisms underlying TSC with epilepsy.
Author Response
Dear Reviewer,
This manuscript employs DTI techniques to characterize differences in brain microstructures among patients with and without epilepsy, as well as in healthy controls. Statistical analyses of DTI parameters across these three groups have highlighted distinctive patterns in regions with impaired white matter. The manuscript is well-written and presents its findings clearly. The manuscript appears to require only minor revisions.
Thank you very much for taking the time to review this manuscript. Please find the detailed responses below and the corresponding revisions/corrections highlighted in the re-submitted files.
Comment 1.: To enhance readability, I recommend providing a more detailed description of TSC with epilepsy in the introduction. Specifically, it would be helpful to introduce the symptoms, patient population and summarize current research on the mechanisms underlying TSC with epilepsy.
Response: Thank you for pointing this out. Although the natural course of epilepsy in TSC is not a main aim of the study we have provided more detailed description of TSC epilepsy in the Introduction (page 2.) We described symptoms, population, mechanisms underlying TSC related epilepsy.
Changes in the structure of the brain, in turn, give rise to the clinical manifestations of epilepsy which is the most prevalent neurological symptom in patients with TSC and affects 70-90% of patients [8,9]. There are different types of convulsions, such as focal, multifocal, infantile, or a combination of these and others. Infantile spasms can occur in up to 30% of people affected by TSC. People with TSC frequently have multiple seizures that do not improve with medication [10]. Epileptogenic foci can occur in different parts of the brain at different times. In longitudinal studies, changes in the morphology of the seizures have been observed in more than half of the patients. The deficiency of GABAergic interneurons explains the early onset and severe course of TSC-related seizures, thus the GABA transaminase inhibitor vigabatrin is proven to be effective in approximately 95% of patients [11]. Other anti-epileptic drugs are less effective in TSC. Some researchers suggest that white matter disruption is related with epilepsy in TSC [12].
Kind regards,
Anna Marcinkowska
Reviewer 3 Report
Comments and Suggestions for Authors
This manuscript investigates the differences in white matter integrity between Tuberous Sclerosis Complex (TSC) patients with and without epilepsy using Diffusion Tensor Imaging (DTI) finding that patients of TSC with epilepsy exhibit more pronounced abnormalities in longer white matter fibers, especially in the left hemisphere, highlighting the potential of DTI as a diagnostic tool in assessing white matter changes in TSC. Before being considered for publication, there are certain issues which need to be addressed properly.
1. The patient recruiting strategy is rather simple and the lack of control over certain parameters could lead to overly simplistic explanations and conclusions that may not be robust.
2. Including a schematic illustration of the detection methods used in studying TSC onset would greatly aid readers in understanding the methodology.
3. The results in Tables 2, 3, and 4 are difficult to interpret because they are presented as raw data rather than being scientifically analyzed and explained.
4. Some of the references cited in the paper are outdated, which could impact the relevance and accuracy of the manuscript. Consider updating the references with more recent studies.
Author Response
Dear Reviewer,
This manuscript investigates the differences in white matter integrity between Tuberous Sclerosis Complex (TSC) patients with and without epilepsy using Diffusion Tensor Imaging (DTI) finding that patients of TSC with epilepsy exhibit more pronounced abnormalities in longer white matter fibers, especially in the left hemisphere, highlighting the potential of DTI as a diagnostic tool in assessing white matter changes in TSC. Before being considered for publication, there are certain issues which need to be addressed properly.
Thank you very much for taking the time to review this manuscript. Please find the detailed responses below and the corresponding revisions/corrections highlighted in the re-submitted files.
Comment 1.: The patient recruiting strategy is rather simple and the lack of control over certain parameters could lead to overly simplistic explanations and conclusions that may not be robust.
Response: Thank you for pointing this out. We agree with this comment. Therefore, we extended description of recruiting process (paragraph 2, page 3 and 4) to avoid misleading conclusions..
All individuals referred to national TSC reference clinical center at the Department of Nephrology, Transplantology and Internal Medicine underwent detailed analysis of medical records, physical examination performed by nephrologist and or internal medicine specialist, skin lesions and oral manifestations assessment, kidneys and brain MRI, and high resolution lungs computed tomography. 100 subjects who met the criteria for a diagnosis of TSC based on the international diagnostic guidelines [40] were examined by clinical psychologist. First qualification for the project was carried out using the TAND Checklist [41]. Further detailed psychological assessment involved mood and cognitive testing as well as assessment of neurodevelopmental disorders [42] for assessing individuals without ASD and intellectual disability.
Comment 2.: Including a schematic illustration of the detection methods used in studying TSC onset would greatly aid readers in understanding the methodology.
Response: Thank you for your suggestion. We have prepared Figure 1. (paragraph 2, page 3) presenting methods used in the study to detect TSC onset in each patient.
Schematic illustration of the detection methods used in studying TSC onset in the study. Before enrollment to the diffusion tensor imaging (DTI) study all patients referred to the TSC reference clnical center underwent detailed assessment performed by nephrologist and/or internal medicine specialist. Medical records were analysed in case of epilepsy in the past and current onsets. Each patient underwent skin and oral assessment for lesions and clinical manifestations of TSC. All patients had kidneys and brain magnetic resonance imaging (MRI) and high resolution comutet tomography (HRCT) of lungs. Patients who met the criteria of TSC were reffered to clinical psychologists examination. All patients were assessed with TAND Checklist. More detailed psychological examination included cognitive and mood testing as well as symptoms of neurodevelopmental disorders.
Comment 3.: The results in Tables 2, 3, and 4 are difficult to interpret because they are presented as raw data rather than being scientifically analyzed and explained.
Response: Thank you for your comment. The tables presented provides a statistical analysis of the DTI parameters for NAWM fibers between the EpiTSC and NEpiTSC groups, rather than raw data. The mean and standard deviation (SD) values for various diffusion metrics such as FA, MD, RD, and AD are calculated from the processed DTI images using statistical methods. These metrics represent averaged values across specific neural pathways, identified and analyzed through the methods described, including the use of probabilistic atlases and advanced image processing techniques. The statistical significance of differences between groups was evaluated using t-tests or U-tests, as indicated by the t/U and p values in the table. (Detailed DTI and statistical analysis are described in paragraph 2.2 and 2.2.1 page 4)
Comment 4.: Some of the references cited in the paper are outdated, which could impact the relevance and accuracy of the manuscript. Consider updating the references with more recent studies.
Response: We have modified the references to emphasize this point. We updated references with more recent studies (pages 10-11).
Kind regards,
Anna Marcinkowska
Round 2
Reviewer 3 Report
Comments and Suggestions for Authors
Most of the issues have been properly addressed